# Beyond Deep Learning: An Evolutionary Feature Engineering Approach to Tabular Data Classification

## Abstract

In recent years, deep learning has achieved impressive performance in the computer vision and natural language processing domains. In the tabular data classification scenario, with the emergence of the transformer architecture, a number of algorithms have been reported to yield better results than conventional tree-based models. Most of these methods attribute the success of deep learning methods to the expressive feature construction capability of neural networks. Nonetheless, in real practice, manually designed high-order features with traditional machine learning methods are still widely used because neural-network-based features can be easy to over-fitting. In this paper, we propose an evolution-based feature engineering algorithm to imitate the manual feature construction process through trial and improvement. Importantly, the evolutionary method provides an opportunity to optimize cross-validation loss, where gradient methods fail to do so. On a large-scale classification benchmark of 119 datasets, the experimental results demonstrate that the proposed method outperforms existing fine-tuned state-of-the-art tree-based and deep-learning-based classification algorithms.

## 1 Introduction

Tabular data is a typical way of storing information in a computer system, and the tabular data learning methodology has been widely employed in recommendation systems (Wang et al., 2021), advertising (Yang et al., 2021), and the medical industry (Spooner et al., 2020) to automate the decision process. Formally, tabular data learning techniques capture the association between some explanatory variables $\{x_1, \ldots, x_m\}$ and a response variable $y$ for a given dataset $\{(\{x_1^1, \ldots, x_m^1\}, y^1), \ldots, (\{x_1^n, \ldots, x_m^n\}, y^n)\}$, where $n$ is the number of data items.

There are various machine learning methods that could address the tabular data learning problem, such as decision trees, linear models, and support vector machines. However, these learning algorithms have a limited learning capability to fully tackle this problem. For example, the linear model assumes that the relationship between explanatory variables and response variables is linear, whereas the decision tree posits that an axis-parallel decision boundary can divide distinct categories of samples. These assumptions are not always correct in the real world, and thus machine learning practitioners always employ feature engineering techniques to construct a set of high-order features $\{\{\phi_1^1 \ldots \phi_k^1\}, \ldots, \{\phi_1^n \ldots \phi_k^n\}\}$ to improve the learning capability of existing algorithms.

In real-world applications, many feature engineering methods can be used by industrial practitioners to construct high-order features before training a machine learning model. The conventional feature engineering strategies include manually designed features, dimensionality reduction methods (Colange et al., 2020), kernel methods (Allen-Zhu & Li, 2019), and so on. However, these methods have drawbacks. Manual feature construction approaches require a large amount of human labor, kernel methods are hard to be combined with tree-based methods, and dimensionality reduction algorithms are insufficient to enhance state-of-the-art learning algorithms, especially for unsupervised ones (Lensen et al., 2022).

With the success of deep learning, there is a growing interest in using neural networks to build high-order features automatically, particularly in the field of recommendation systems (Song et al., 2019; Lian et al., 2018). However, it is debatable whether deep learning techniques can learn high-order features effectively (Wang et al., 2021). Despite the fact that various neural network-based

tabular learning algorithms have shown better predictive accuracy than some tree-based ensemble models (Arık & Pfister, 2021), most of these methods only validate their algorithms on datasets with a large number of training samples and homogeneous features. Nowadays, it is still controversial whether deep learning methods outperform tree-based methods on datasets with heterogeneous features and a limited amount of data (Gorishniy et al., 2021), and we hope to further explore this problem in this paper.

In real-world cases, machine learning engineers typically start from some randomly designed features, and then evaluate these features to get cross-validation loss and feature importance value. The feedback information can be used by engineers to design better features. To imitate this workflow, we design a feature engineering method named evolutionary feature engineering automation tool (EvoFeat)[1], which constructs nonlinear features for enhancing an ensemble of decision tree and logistic regression models, and thus achieving better results than conventional ensemble learning methods. The three objectives of our paper are summarized as follows:

- Considering cross-validation losses and feature importance values are widely used by machine learning engineers in the feature engineering process, we propose an evolutionary feature construction method that uses both information to guide the search process.
- Given that the performance of the constructed features is highly bonded to the base learner, we propose a mechanism that allows the evolutionary algorithm to automatically determine whether to use random decision trees or logistic regression models as base learners.
- We conduct experiments on 119 datasets and compare our method to six traditional machine learning algorithms, four deep learning algorithms and two automated feature construction methods. Experiment results show that deep learning methods are not silver bullets for classification tasks.

The remainder of the paper is structured as follows. Section 2 introduces related work of feature construction. Section 3 discusses the motivation behind our work. Section 4 then explains the details of the proposed method, followed by the experimental results provided in Section 5. Finally, Section 6 summarizes our paper and recommends some intriguing future avenues.

## 2 RELATED WORK

Evolutionary feature construction methods have received a lot of attention in the evolutionary learning domain. Nonetheless, the majority of these methods focus on simple learners, such as a single decision tree (Tran et al., 2019) or a single support vector machine (Nag & Pal, 2019). Research on how to apply evolutionary feature construction methods to improve state-of-the-art classification algorithms is still lacking. In recent years, ensemble-based evolutionary feature construction methods have produced encouraging results on both the symbolic regression (Zhang et al., 2021) and image classification (Bi et al., 2020; 2021a) tasks. However, feature construction on these tasks are different than tabular data classification tasks because domain knowledge for designing primitive functions are available for these tasks, such as a function named analytical quotient Ni et al. (2012) for symbolic regression and the histogram of oriented gradients (HOG) Dalal & Triggs (2005) for image classification.

In addition to the evolution-based method, beam search could also be used to construct high-order features (Katz et al., 2016; Luo et al., 2019; Shi et al., 2020). The main idea behind these methods is to start with some low-order features and iteratively generate higher-order features based on important low-order features. A meta learner (Katz et al., 2016), logistic regression accuracy (Luo et al., 2019), or XGBoost feature importance (Shi et al., 2020) can be used to determine the feature importance of each generated feature. However, these methods lack an effective mechanism to prevent over-fitting and thus may have restricted feature construction capabilities.

In the recommendation system domain, deep learning methods have been shown to be an effective way to construct high-order features for tabular data classification tasks (Guo et al., 2017). Even though the feed-forward neural network (FNN) is considered a universal function approximator (Kidger & Lyons, 2020), its ability to model high-order feature interactions has been questioned (Mudgal et al., 2018). In order to efficiently construct high-order features, a cross network is designed in Deep & Cross Network (DCN) to explicitly model the feature interaction (Mudgal et al., 2018).

---

[1]Source code: `https://tinyurl.com/ICLR2023-EvoFEAT`

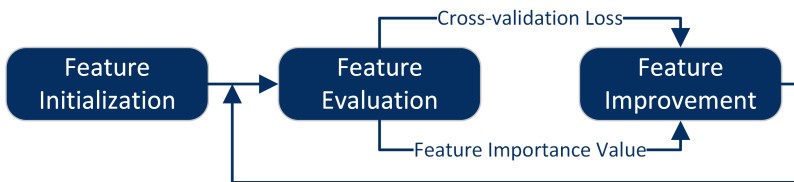

Figure 1: An example of the feature engineering process.

Following that, xDeepFM employs a field-wise feature cross method to effectively (Lian et al., 2018) model the feature interaction in a bit-wise manner. AutoInt (Song et al., 2019) goes a step further, learning feature interaction using the attention mechanism rather than a simple outer product. Due to useless feature interaction can introduce noise, AutoFIS (Liu et al., 2020) employs a generalized regularized dual averaging (GRDA) (Chao & Cheng, 2019) method to remove unimportant feature interaction. With the increasing popularity of the neural network architecture search (NAS) technique, an algorithm known as AutoPI (Meng et al., 2021) employs the NAS technique to automatically discover better feature interaction components. This method is inspired by differentiable architecture search (DARTS) (Liu et al., 2018), and it uses the temperature anneal (Liu et al., 2018) technique to find the best subnet from a supernet using the gradient descent method. However, the effectiveness of these feature cross modules over FNN is still debatable (Wang et al., 2021), let alone the lack of a comprehensive study of whether these approaches can outperform XGBoost on classical classification problems (Gorishniy et al., 2021).

## 3 PRELIMINARIES

First of all, let us analyze how a machine learning engineer constructs features for a given dataset $\{(\{x_1^1, \ldots, x_m^1\}, y^1), \ldots, (\{x_1^n, \ldots, x_m^n\}, y^n)\}$ in the absence of domain expertise. Figure 1 depicts an example of the feature engineering workflow. It is divided into three phases:

- **Feature Initialization**: First, a machine learning engineer without any domain knowledge can randomly construct some features $\Phi = \{\phi_1 \ldots \phi_k\}$ based on original features $\{x_1 \ldots x_m\}$.
- **Feature Evaluation**: Second, using the cross-validation approach, the generated features $\{\phi_1 \ldots \phi_k\}$ are evaluated on the training set and the target learning algorithm. The trained model produces cross-validation losses and feature importance values to help an engineer determine useful features.
- **Feature Improvement**: Finally, the engineer removes some useless features and replaces them with new features derived from important features.

During the feature construction process, the second and third phases are repeated until the engineer is satisfied with the result. Formally, if $F$ is a search space of machine learning models and $L(f(x), y)$ is the loss function of a model $f \in F$, the feature construction process is to find a set of features $\Phi$ that can minimize the objective function $\min_{f^* \in F} L(f^*(\Phi(x)), y)$.

In real-world applications, some machine learning engineers prefer to only use feature importance values to detect useful features (Shi et al., 2020). However, this is risky because the feature importance values do not reflect the generalization performance of the constructed features. In other words, a feature may play an important role in an over-fitted model, and we must have a strategy to avoid sticking at this local optima. Constructing numerous sets of features at the same time is a plausible idea. The engineer can then detect the over-fitting issue based on the cross-validation performance of multiple sets of features and make a more informed decision.

The feature engineering paradigm depicted in Figure 1 is general. Even for a machine learning algorithm that does not provide an internal feature importance report, it can still be employed in this feature engineering method based on some black-box feature importance calculation approaches, such as SHAP (Lundberg & Lee, 2017). Under this paradigm, the most critical problem is to determine how to leverage information based on feature importance values and cross-validation loss to generate better features. Here, we employ tree-based genetic programming (TGP) to do so. TGP is an evolutionary algorithm that focuses on evolving symbolic trees. Its gradient free nature and variable length representation that make it suitable to generate expressive features based on non-differentiable objective values.

## 4 The Proposed Algorithm

### 4.1 Feature Representation

First of all, unlike the traditional evolutionary algorithm, which utilizes an array to denote a solution, we employ a set of symbolic trees to represent a set of constructed features. Each symbolic tree consists of two parts: non-leaf nodes and leaf nodes. Non-leaf nodes are also known as functions, which can transform or combine basic features, such as $\{+, -\}$ or $\{log, sin\}$. The leaf nodes are also known as terminals, representing original features, such as $\{x_1, x_2\}$. Each symbolic tree can be used to construct a feature $\phi$, and all symbolic trees in an individual $\Phi$ together could form a new feature space $\mathbb{S}$. In addition to a set of symbolic trees, each individual is also accompanied by a base learner, which is a decision tree or a linear regression model in this research. This base learner is used to evaluate the predictive performance of each feature space and also used to give a prediction of unseen data.

### 4.2 Algorithm Framework

The proposed evolutionary algorithm consists of five parts: initialization, evaluation, selection, generation, and archive update. The pseudocode for our algorithm is shown in Algorithm 1.

**Initialization:** In the initialization stage, the algorithm randomly initializes $N$ individuals $\{\Phi_1 \ldots \Phi_N\}$ to form the initial population. Each individual consists of $k$ symbolic trees $\{\phi_1 \ldots \phi_k\}$, which could be used to construct a $k$-dimensional feature space.

**Evaluation:** At the evaluation stage, we construct training data based on each individual $\Phi$, and evaluate them on their corresponding base learners $f$. The fitness value of each individual can be calculated using the five-fold cross-validation loss, and the fitness value of each feature can be obtained using the average feature importance values across five folds.

**Selection:** The selection operator selects promising individuals $\{\Phi_A, \Phi_B\}$ from the population $P$ for crossover and mutation. In this paper, we use the lexicase selection operator (La Cava et al., 2019) to select parent individuals based on the cross-validation losses.

**Generation:** The generation operator generates new individuals based on selected parent features. We use self-competitive based crossover (Nguyen et al., 2021) and guided mutation operators to increase search efficiency by leveraging feature importance values.

**Archive Update:** At the end of each iteration, we dynamically update an archive $A$ to store historically top-performing models, depending on the performance of archived models $A$ and newly generated models $\{f_1 \ldots f_N\}$. The archive update method employed in this research is a greedy ensemble selection method named reduce-error pruning (Caruana et al., 2004).

The selection, generation, evaluation and archive update operators are executed repeatedly until the maximum number of generations is reached. Finally, all base learners in archive $A$ could form an ensemble model based on the weights assigned by the ensemble selection method.

### 4.3 Feature Initialization

The proposed evolutionary feature construction method is based on TGP, and thus we use a ramped-half-and-half (Burke et al., 2003) symbolic tree initialization strategy from TGP to initialize each tree $\phi$. In particular, half of the symbolic trees are built as full symbolic trees, while the other half are initialized with a random height. In addition, each individual is randomly assigned a random decision tree (RDT) or logistic regression model (LR) with equal probability. This allows the evolutionary algorithm to determine the optimal base learner to form an ensemble model.

### 4.4 Feature Selection

The main purpose of the feature selection operator is to select two individuals from the population based on the cross-validation loss. In the existing EA literature, numerous selection operators are available, such as age-fitness pareto (AFP) (Schmidt & Lipson, 2011), double tournament selection (Luke & Panait, 2002), lexicase selection (La Cava et al., 2019), and so on. In this research, we employ lexicase selection as the selection operator because it is helpful to conserve population diversity, which is advantageous in the ensemble learning scenario (Zhang et al., 2021).

**Lexicase Selection:** For a given vector of validation losses, the lexicase selection operator constructs a set of filters based on each loss value across all individuals. Each filter randomly selects one dimension

---

**Algorithm 1** Evolutionary Feature Engineering Automation Tool (EvoFeat)

---

**Input:** Population Size $N$, Number of Features $k$, Number of Generations $max\_gen$
**Output:** Archive $A$
1: Randomly initialize $N$ set of features $P = \{\Phi_1 \dots \Phi_N\}$        ▷ Feature Initialization
2: **for** $i = 1, \dots N$ **do**        ▷ Feature Evaluation
3:      $f_i \leftarrow$ cross-validation$(f_i, \Phi_i)$
4: $A \leftarrow \emptyset$
5: $A \leftarrow$ archive update $(\{f_1 \dots f_N\}, A)$        ▷ Archive Update
6: $gen \leftarrow 0$
7: **while** $gen < max\_gen$ **do**        ▷ Main loop
8:      $P' = \emptyset$
9:      **for** $i = 1, \cdots, N$ **do**
10:          $P_{LR}, P_{DT} \leftarrow$ population division$(P)$
11:          **if** $rand() < 0.25$ **then**        ▷ Base Learner Selection
12:              $\Phi_A \leftarrow$ lexicase selection$(P_{LR})$
13:              $\Phi_B \leftarrow$ lexicase selection$(P_{DT})$
14:          **else**
15:              $P' \leftarrow$ random selection$(P_{LR}, P_{DT})$
16:              $\Phi_A, \Phi_B \leftarrow$ lexicase selection$(P')$
       ▷ Individual Selection
17:          **for** $j = 1, \cdots, \lfloor\frac{k}{2}\rfloor$ **do**        ▷ Crossover (Feature Generation)
18:              **if** $rand() < p_c$ **then**
19:                  $\phi_a, \phi_b \leftarrow$ self-competitive selection$(\Phi_A)$        ▷ Feature Selection
20:                  $\phi_c, \phi_d \leftarrow$ self-competitive selection$(\Phi_B)$
21:                  $\phi_a \leftarrow$ crossover$(\phi_a, \phi_d)$
22:                  $\phi_c \leftarrow$ crossover$(\phi_c, \phi_b)$
23:          **for** $j = 1, \cdots, k$ **do**        ▷ Mutation (Feature Generation)
24:              **if** $rand() < p_m$ **then**
25:                  $\phi_a \leftarrow$ random selection$(\Phi_A)$
26:                  $\phi_i \leftarrow$ mutation$(\phi_i)$
27:          $P' = P' \cup \{\Phi_A, \Phi_B\}$
28:      $P = P'$
29:      **for** $k = 1, \dots N$ **do**        ▷ Feature Evaluation
30:          $f_k \leftarrow$ cross-validation$(f_i, \Phi_k)$
31:      $A \leftarrow$ archive update $(\{f_1 \dots f_N\}, A)$        ▷ Archive Update
32:      $gen \leftarrow gen + 1$
33: **return** $A$

---

$j \in [1, n]$ to do filtering, and the threshold $\tau_j$ is dynamically set as $\tau_j = \min_i mse_j^i + mad_j$, where $mad_j = \underset{i}{median}(|mse_j^i - \underset{k}{median}(mse_j^k)|)$ represents the median absolute deviation. Based on this selection operator, we can select a suitable number of parent individuals for crossover and selection.

**Base Learner Selection:** We employ either a decision tree or logistic regression model as the base learner to evaluate each group of features. However, decision trees and logistic regression models may not produce comparable predictive results. For example, because a single decision tree cannot produce a categorical distribution, misclassifying only one example will result in an infinite cross-entropy loss. Thus, we borrow an idea from the multitask evolutionary optimization domain (Wei et al., 2021) to select two parent individuals from a heterogeneous population. First, we divide the entire population into two subgroups: one with individuals exclusively using decision trees, and the other with individuals only using logistic regression. Then we define a hyperparameter called random mating probability $rmp = 0.25$, which signifies the probability of selecting two parents from two subgroups. In the remaining seventy five percent of cases, we select two parents from either subgroup.

### 4.5 FEATURE GENERATION

There are two operators for transforming two given parent individuals into new individuals, namely a self-competitive crossover operator and a guided mutation operator. The self-competitive crossover

operator was originally developed to evolve multiple job scheduling rules (Nguyen et al., 2021). It keeps effective rules and modifies the less effective rules, which imitates the manual rule design process and is suitable for our feature construction scenario. Guided mutation was proposed to solve maximum clique problems (Zhang et al., 2005) by eliminating less useful variables, which is also important for classification problem. Thus, we decide to use these two operators in our algorithm.

**Self-Competitive Crossover:** First, we utilize the self-competitive crossover operator to exchange beneficial materials of two parent individuals. In general, the goal of the self-competitive operator is to move important materials from the best feature to the worst feature while keeping the best feature intact. For a given set of feature importance values $\{\theta_1 \ldots \theta_k\}$ provided by the base learner, we can use a softmax function, $P(\theta_i) = \frac{e^{\theta_i/T}}{\sum_{j=1}^{k} e^{\theta_j/T}}$, where $T$ represents temperature, to determine the likelihood of selecting each feature $\theta_i$ as the donor, and use a negative softmax function to determine the acceptor.

Given the preceding probability distribution, we begin by sampling the acceptor individual $\Phi_{acceptor}$ negatively proportionate to $P(\theta_i)$, i.e., $P(-\theta_i)$. Then we sample another individual $\Phi_{donor}$ as the donor individual according to the probability $P(\theta_i)$. Thereafter, the crossover operator selects a subtree from $\Phi_{donor}$ at random to replace a subtree of $\Phi_{acceptor}$. This crossover operator ensures that the top-performing features are rarely destroyed, while the worst features have a high probability of being improved.

**Guided Mutation:** In addition to the self-competitive operator, the guided mutation operator is utilized to generate new features. The guided mutation operator is similar to the random mutation operator in that it replaces a subtree of a parent individual with a randomly generated subtree. However, the guided mutation operator uses a guided probability vector $p$ to determine the probability of using each terminal variable in a newly generated subtree.

For each terminal variable $x$, the guided probability $\{p_1 \ldots p_m\}$ is defined as the percentage of occurrences of variable $x$ in all features in the archive, weighted by the feature importance of each feature. The feature importance values of a decision tree are calculated by counting the total reduction of Gini impurity with each feature $\phi$, whereas the feature importance values in a linear model are obtained by calculating the absolute value of model coefficients. All features are standardized before constructing the linear model. This ensures that the magnitude of each feature will not have an impact on the model coefficient.

### 4.6 Feature Evaluation

For each group of features, feature evaluation is conducted on its corresponding classification model. To avoid over-fitting, we validate the effectiveness of each set of features using a cross-validation approach. For any given training set, we partition it into five subsets, i.e., $\{X_1 \ldots X_5\}$. Then, for each round, we utilize four folds to train a model and the remaining folds to validate the learned model.

Instead of summing all losses into a scalar, we keep the original loss vector because it contains more detailed and useful information which the selection operator can exploit and utilize. For each training instance, we employ cross entropy as the fitness function. If the problem contains $C$ classes, the cross entropy loss can be defined as $\sum_{c \in C} p_c * log(q_c)$, where $q_c$ refers to the probability of the model prediction belonging to a certain class $c$ as $q_c$ and the ground truth probability distribution is $p_c$.

## 5 Experiments

We conduct extensive experiments on synthetic and real-world datasets to answer the following three questions:

- **Q1**: How does the proposed method perform compared with traditional machine learning (ML) methods (Section 5.2.1), deep learning-based (DL) methods (Section 5.2.2), and open-source feature engineering (FE) tools (Section 5.2.3)?
- **Q2**: How does dynamically selecting base learners during the evolutionary process perform compared with determining them in advance (Section 5.3.1)?
- **Q3**: Is it more efficient to use both cross-validation losses and feature importance values to guide the search than using either one alone (Section 5.3.2)?

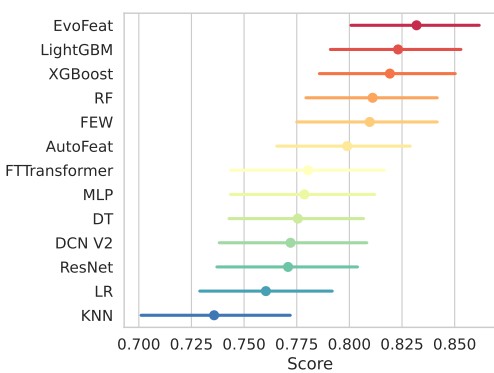

| Algorithm | Mean | Imp. ↑ |
|---|---|---|
| EvoFeat⋆ | 0.831899 | - |
| LightGBM† | 0.823112 | 1.07% |
| XGBoost† | 0.819236 | 1.55% |
| RF† | 0.811038 | 2.57% |
| FEW◇ | 0.809664 | 2.75% |
| AutoFeat◇ | 0.798939 | 4.13% |
| FTTransformer‡ | 0.780356 | 6.60% |
| MLP‡ | 0.778617 | 6.84% |
| DT† | 0.775535 | 7.27% |
| DCN V2‡ | 0.772064 | 7.75% |
| ResNet‡ | 0.770875 | 7.92% |
| LR† | 0.760355 | 9.41% |
| KNN† | 0.735761 | 13.07% |

Figure 2: Comparison of balanced testing accuracy distributions over different algorithms on 119 PMLB dataset.

Table 1: Statistical results of balanced testing accuracy for different algorithms on 119 PMLB datasets. (⋆: Our algorithm, †: ML algorithms, ‡: DL algorithms, ◇: FE algorithms, Imp. ↑: Relative improvement)

## 5.1 EXPERIMENTAL SETTINGS

**Experimental Datasets:** The experimental datasets include all datasets in DIGEN (Orzechowski & Moore, 2021)[2] and all PMLB (Romano et al., 2021) [3] classification datasets between 200 and 10000 instances. DIGEN is a diverse set of datasets synthesized by the genetic programming method, whereas PMLB includes real-world datasets collected from OpenML (Vanschoren et al., 2014). In large-scale experiments, we test the performance of our approach against other machine learning algorithms on all 119 datasets in PMLB with up to 10000 instances. In ablation studies, we use all datasets in DIGEN and all datasets in PMLB ranging from 200 to 2000 instances due to limited computational resources.

**Evaluation Protocol:** For the evaluation protocol, we randomly sample 80% data as the training set to train a model and use the remaining 20% data as the test set to calculate the testing balanced accuracy of the trained model. To obtain a reliable result, we repeat all experiments with ten random seeds and take the average score as the final score. For all baseline algorithms, we tune them by Heteroscedastic Evolutionary Bayesian Optimization (HEBO) (Cowen-Rivers et al., 2020) or heuristic methods. The details of the tuning methods and all hyperparameter search spaces are shown in the supplementary material.

## 5.2 LARGE-SCALE EXPERIMENT

### 5.2.1 COMPARISON WITH TRADITIONAL MACHINE LEARNING METHODS

**Baseline Algorithms:** First of all, we compare our algorithm with traditional machine learning algorithms to demonstrate how feature construction improves the performance. We compare our algorithm to six traditional machine learning classification algorithms, including XGBoost (Chen & Guestrin, 2016), LightGBM (Ke et al., 2017), Random Forest (RF), Decision Tree (DT), Logistic Regression (LR), and K-Nearest Neighbors (KNN). Among these methods, XGBoost and LightGBM are regarded as state-of-the-art algorithms (Gorishniy et al., 2021).

**Experimental Results:** Figure 2 illustrates the testing accuracy distribution of all algorithms across 119 datasets. It is clear that gradient boosting methods, such as XGBoost and LightGBM, are top-performing classification algorithms. However, our approach achieves a better average accuracy than these algorithms. The numerical results shown in Table 1 confirm this, as our method achieves an average accuracy of 83%, which has more than 1% improvement over the average accuracies of XGBoost and LightGBM. A further analysis shows that EvoFeat ranks first on 51 out of 119 datasets,

---

[2]https://github.com/EpistasisLab/digen
[3]https://github.com/EpistasisLab/pmlb

while LightGBM only ranks first on 26 datasets. These results demonstrate that the evolution-based feature construction method enables an ensemble learning model composed of simple base learners to outperform state-of-the-art machine learning models. It is worth mentioning that EvoFeat ranks first does not mean it performs the best on every dataset. Due to space constraints, we discuss this topic in the supplementary material using the instance space analysis technique.

### 5.2.2 COMPARISON WITH DEEP LEARNING METHODS

**Baseline Algorithms:** There are a lot of deep learning-based methods that have been proposed in recent years for tabular data learning. We select four models from the existing literature for comparison, i.e., multilayer perceptron (MLP), ResNet, DCN V2 (Wang et al., 2021) and FT-Transformer (Gorishniy et al., 2021). The first two architectures are commonly used baselines in the existing literature. The third one is the state-of-the-art model for click-through rate (CTR) prediction, and the fourth model is the state-of-the-art model for tabular data learning.

**Experimental Results:** The experimental results in Table 1 reveal that existing deep learning approaches cannot compete with XGBoost or LightGBM because the best one, FT-Transformer, only achieves 78% average accuracy. Surprisingly, all deep learning algorithms perform similarly to a fine-tuned decision tree, which has an average accuracy of 77%. One probable explanation is that these deep learning algorithms require a large amount of training data and thus fail to learn a robust feature representation on datasets with less than ten thousand items, which is however typical for tabular data.

### 5.2.3 COMPARISON WITH FEATURE CONSTRUCTION METHODS

**Baseline Algorithms:** In this paper, we compare our method to two open-source feature construction methods: Feature engineering wrapper (FEW) (La Cava & Moore, 2017b) and AutoFeat (Horn et al., 2019). FEW is a genetic programming based feature construction method, whereas AutoFeat is an exhaustive search based feature engineering method. They are used to construct features for the best model among Extra Trees (ET), RF, XGBoost and LightGBM. The details of experiment settings are given in the supplementary material.

**Experimental Results:** Experiment results in Table 1 show that both FEW and AutoFeat cannot achieve top performance, with accuracy scores of 80% and 79%, respectively. FEW uses the coefficient of determination and feature importance values to search for high-order features, and uses the cross-validation loss to determine the historically best set of features. However, generating features without considering the cross-validation loss may mislead the search process and diminish the likelihood of uncovering expressive features. AutoFeat employs L1-regularized LR to determine superfluous high-order features. However, features useful on LR may not work well on RF, and therefore its feature construction capability is also limited.

### 5.3 ABLATION STUDIES

### 5.3.1 BASE LEARNERS

Due to the difficulties of generating a diverse set of linear models, most ensemble learning systems employ decision trees as base learners rather than linear regression models. However, the construction of non-linear features may lead to different results. Therefore, we perform an ablation study in which we only use linear regression models or decision trees during the evolutionary feature construction process to investigate the benefit of simultaneously constructing features for linear regression models and decision trees.

Figure 3 illustrates the balanced testing accuracy with various base learners over 20 datasets in PMLB. The experimental results reveal that there is no free lunch when it comes to addressing classification problems, and each method has its own set of niching tasks. Nonetheless, the combination of random decision tree and logistic regression allows our system to automatically select the appropriate model on the fly, resulting in improved average performance. Table 2 shows the mean accuracy of different base learners over 89 PMLB datasets and 40 DIGEN datasets. This table illustrates that the combination of DT and LR achieves an accuracy score of 86%, which is more than 1.2% higher than dropping either of them, confirming the effectiveness of using heterogeneous base learners.

### 5.3.2 EVOLUTIONARY ALGORITHMS

In this paper, we propose to use an evolutionary approach to exploit the information provided by feature importance and cross-validation loss. It is worthwhile investigating whether the evolutionary

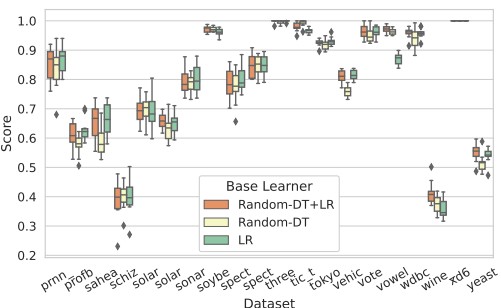

Figure 3: The balanced testing accuracy of our algorithm with respect to different base learners.

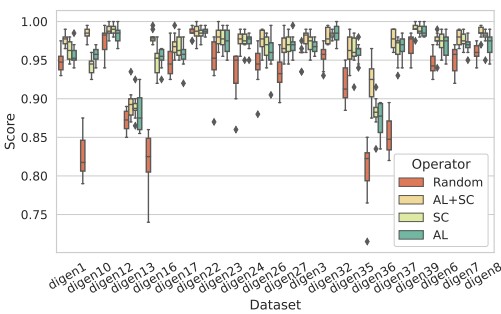

Figure 4: The balanced testing accuracy of our algorithm with respect to different selection operators.

| Base Learner | Mean | Imp. ↑ |
|---|---|---|
| Random-DT+LR | 0.860044 | - |
| LR | 0.849455 | 1.25% |
| Random-DT | 0.848104 | 1.41% |

Table 2: Comparison of balanced testing accuracy over different base learners.

| Selection Operator | Mean | Imp. ↑ |
|---|---|---|
| AL+SC | 0.860044 | - |
| AL | 0.857332 | 0.32% |
| SC | 0.855892 | 0.49% |
| Random | 0.841332 | 2.22% |

Table 3: Comparison of balanced testing accuracy over different selection operators.

algorithm effectively uses this information during the search process. Otherwise, random search would be a preferable option because it is much simpler. Therefore, this section presents an ablation experiment to demonstrate the efficacy of combined selection operators, which is the driving force of an evolutionary algorithm.

Four selection operators: random selection, only automatic lexicase selection (AL), only self-competitive selection (SC), and both AL and SC (AL+SC) are compared in this section. Figure 4 presents the distribution of testing accuracy with respect to three selection mechanisms on 20 DIGEN datasets. From this figure, it is clear that using two selection operators simultaneously can effectively improve the predictive accuracy of the final searched features. Table 3 displays the mean testing classification accuracy across 129 datasets for different selection operators. These results confirm the effectiveness of using both selection operators in the searching process, where the accuracy score of combining two operators is more than 0.32% higher than that of using only one of these operators. Therefore, we can conclude that leveraging both cross-validation loss and feature-importance information is beneficial for obtaining better final results.

# 6 CONCLUSIONS

Automated feature engineering is a critical component of an automated machine learning system. In this research, we develop and present an evolution-based feature construction method for augmenting standard machine learning algorithms. To make the technique applicable to a wide range of tasks, we simultaneously evolve features for decision trees and linear regression models and form an ensemble of top-performing models. To increase the search capability of our algorithm, we use both feature importance values and cross-validation losses to identify promising features. Experimental results on 119 datasets show that the proposed method outperforms both state-of-the-art tree-based ensemble learning methods and deep learning methods.

For future study, considering the current base learner is confined to decision trees and logistic regression models, it is worth investigating whether other learners, such as neural networks or GBDTs, could be added to this framework. Furthermore, exploring how to adapt our algorithm to large-scale datasets is an intriguing issue; instance selection methods (Bi et al., 2021b) may be required to reduce computational complexity.

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
