# OpenReview forum: "Beyond Deep Learning: An Evolutionary Feature Engineering Approach to Tabular Data Classification"
_ICLR.cc/2023/Conference — Submitted to ICLR 2023_

### Official Review · Reviewer_LFDt · 2022-10-21

**Confidence:** 3
**Correctness:** 3
**Technical Novelty And Significance:** 3
**Empirical Novelty And Significance:** 3
**Recommendation:** 6

**Clarity, Quality, Novelty And Reproducibility:**

- The paper is quite clear, although it requires acquaintance with cited papers to understand what was done. The runtime measurements are not a main aspect of the results, but they are not fully clear (e.g., did each model run on a separate server, did all the servers have identical RAM, etc.).
- It does represent a novel method to use this type of feature engineering for tabular data classification.
- The provided code should enable reproducibility and the data is public. A question I have about the code is regarding the tests, which seem to relate to regression rather than classification. What is the reason for this? How was classification tested and how should it be run?

**Strength And Weaknesses:**

The paper presents a new method for an important problem, and it is reasonably clear despite the space constraints. I think it would be interesting to see how sensitive are the results to the hyperparameter selection of the baselines compared. Specifically, what would be the results of the two top baselines (LightGBM and XGBoost) when simply using their default hyper-parameters? What would happen if the maximal number of estimators in the hyper-parameter search is allowed to be 5000 instead of 1000?
Also, the runtime measurements are not a main aspect of the results, but they are not fully clear (e.g., did each model run on a separate server, did all the servers have identical RAM, etc.).

**Summary Of The Paper:**

The paper considers the feature engineering problem for tabular data classification. It presents a new evolutionary algorithm for performing feature engineering. Each feature is generated by a feature tree, whose leaves are features and whose internal nodes are operators. The number of features is a hyperparameter. The evolutionary search process, finding those trees, is guided by the feature importance and training losses, computed by two base learners: logistic regression or decision tree. The best models at each stage are stored, and the final answer is their ensemble. Ablation study shows that considering both logistic regression and decision-tree is better than using just one of them.
The new method is applied to 119 PMLB datasets (with up to 10K samples each), and compared to two previous feature engineering methods, six classical ML methods and 4 deep-learning methods. The proposed method out performs all previous methods, with an advantage of about 0.8% average accuracy over the next best model, LightGBM. Previous work considered evolutionary algorithms for regression or image classification, but not for tabular data classification.


**Summary Of The Review:**

This is an interesting paper that suggests a new method for an important problem. My main concern is whether the accuracy improvement could be due to non-optimal selection of hyper-parameters for the baselines. I suggest to try the default hyper-parameters for the two top baselines (XGBoost and LightGBM), and also try a hyperparameter search with more estimators (up to 5000). I also would like to have a clarification about the run-time measurements, as I mentioned above, and about my question above about the code. I will update my recommendation based on the author responses.

---

> ### Author Response · Authors · 2022-11-15
> **Response to Reviewer LFDt (Part I)**
>
> **Q1: My main concern is whether the accuracy improvement could be due to non-optimal selection of hyper-parameters for the baselines. I suggest to try the default hyper-parameters for the two top baselines (XGBoost and LightGBM), and also try a hyperparameter search with more estimators (up to 5000).**
>
> A: Thanks. We add more experiments according to your suggestions. Experimental results are presented as follows. "LightGBM(5000)" and "XGBoost(5000)" represent using the hyperparameter space according to your suggestion. However, it seems that using up to 5000 estimators will make the Bayesian optimization algorithm unable to find good enough solutions in a limited time, and results in worse performance than using the settings in our original paper. Also, the accuracy of using the default hyperparameters is inferior to the accuracy of using the tuned hyperparameters.
>
> | Algorithm         |    Score | Improvement |
> | :-: | :-: | :-: |
> | EvoFeat           |            0.831899 | -                     |
> | LightGBM          |            0.823112 | 1.07%                |
> | LightGBM(5000)    |            0.821251 | 1.30%                |
> | XGBoost           |            0.819236 | 1.55%                |
> | XGBoost(5000)     |            0.813254 | 2.29%                |
> | RF                |            0.811038 | 2.57%                |
> | FEW(Default)      |            0.809664 | 2.75%                |
> | AutoFeat(Default) |            0.798939 | 4.13%                |
> | LightGBM(Default) |            0.796962 | 4.38%                |
> | XGBoost(Default)  |            0.796155 | 4.49%                |
> | FTTransformer     |            0.780356 | 6.60%                |
> | MLP               |            0.778617 | 6.84%                |
> | DT                |            0.775535 | 7.27%                |
> | DCN V2            |            0.772064 | 7.75%                |
> | ResNet            |            0.770875 | 7.92%                |
> | LR                |            0.760355 | 9.41%                |
> | KNN               |            0.735761 | 13.07%               |

---

> > ### Comment · Reviewer_LFDt · 2022-11-20
> > **Thank you for your answers**
> >
> > Thank you for your answers and additional experimental results, I will raise my score.

---

> ### Author Response · Authors · 2022-11-15
> **Response to Reviewer LFDt (Part II)**
>
> **Q2: The paper is quite clear, although it requires acquaintance with cited papers to understand what was done. The runtime measurements are not a main aspect of the results, but they are not fully clear (e.g., did each model run on a separate server, did all the servers have identical RAM, etc.).**
>
> A: Thanks. For the runtime measurements, all the experiments are conducted on the same single server with identical CPU and RAM. We have revised our paper to state it clearly.
>
> **Q3: The provided code should enable reproducibility and the data is public. A question I have about the code is regarding the tests, which seem to relate to regression rather than classification. What is the reason for this? How was classification tested and how should it be run?**
>
> A: Thanks. The provided link in our paper is a classifier, which is a minimal example to train a EvoFeat classifier on the Iris dataset. Our code is based on an open-source package for regression tasks. However, we made modifications on base learners (Regression Tree->Logistic Regression+Classification Tree), loss function (MSE->Cross Entropy) and ensemble strategy (Averaging of prediction->Averaging of probability estimation) to make it support classification.

---

### Official Review · Reviewer_bBog · 2022-10-24

**Confidence:** 3
**Clarity, Quality, Novelty And Reproducibility:** This paper cannot bring any new techn…
**Correctness:** 3
**Technical Novelty And Significance:** 2
**Empirical Novelty And Significance:** 2
**Recommendation:** 3

**Strength And Weaknesses:**

Strength:

S1. The author illustrates that in the tabular data classification, the selected feature based on deep learning is prone to overfitting, and the proposed method based on the evolutionary algorithm can obtain more robust and better results.
S2. The proposed method demonstrates promising results compared with various baselines (e.g., ML-based, DL-based, ...).

Weaknesses:

W1. The technical novelty of this paper is limited. There is no novel technique has been proposed or discussed in this paper.
W2. In the real world, tabular data is usually very large-scale, and as the authors say in Section 5.1, they use part of the data due to limited computational resources. Therefore, an efficiency analysis (e.g., execution time, memory) of the proposed method is necessary, and a comparison with other baselines can also be added to comprehensively evaluate the applicability of each method.
W3.The insights for the proposed method are not clear. It would be better to provide a more explicit motivation and vital support for the proposed approach than to beat some approaches only.

**Summary Of The Paper:**

This paper points out that neural-network-based features can be easy to over-fitting and proposes a an evolution-based feature engineering algorithm to imitate the manual feature construction process through trial and improvement.

**Summary Of The Review:**

This paper focus on an interesting problem and gives some experimental analysis. However, The technical novelty of this paper is limited.

---

> ### Author Response · Authors · 2022-11-15
> **Response to Reviewer bBog (Part I)**
>
> **Q1: The technical novelty of this paper is limited. There is no novel technique has been proposed or discussed in this paper.**
>
> A: Thanks. There are two novel contributions to this paper: a two-layer selection paradigm and a multitask optimization framework. The first contribution is about using both feature importance value calculation and cross-validation loss calculation to search for optimal features. Although such an idea seems to be intuitive, our new experimental results in the supplementary material show that using a selection operator based on feature importance values will only work well if we use a SoftMax function with a low temperature to preprocess importance values to fully exploit important features. Also, experimental results show that the selection operator based on cross-validation losses will only work well if we consider the error vector instead of the aggregated errors. We believe these are important findings for the community.
>
> Moreover, our framework co-evolves features and machine learning models under a multitask optimization framework. Beyond the benefit of selecting optimal classifiers on the fly, another advantage of our paradigm is that transferring features from one classifier (DT/LR) to another classifier (LR/DT) is helpful to avoid being stuck at the local optima in some cases. The following are the experimental results that show the fitness values of offspring generated by crossing two homogenous individuals or transferring subtrees of one kind of individuals to another kind of individuals. The results show that the transfer optimization paradigm is beneficial in 3 out of the 20 cases.
>
> |              | DT     | LR     | DT+LR->DT   | DT+LR->LR   |
> |:-:|:-:|:-:|:-:|:-:|
> | digen1_6265  | 0.775  | 0.9164 | 0.771(=)    | 0.9139(=)   |
> | digen2_6949  | 0.5864 | 0.6627 | 0.6111(+)   | 0.657(=)    |
> | digen3_769   | 0.7878 | 0.9251 | 0.7911(=)   | 0.9265(=)   |
> | digen4_860   | 0.5877 | 0.5513 | 0.5858(=)   | 0.551(=)    |
> | digen5_6949  | 0.7829 | 0.7159 | 0.7776(=)   | 0.727(+)    |
> | digen6_466   | 0.6478 | 0.6524 | 0.6393(=)   | 0.6502(=)   |
> | digen7_6949  | 0.6055 | 0.9221 | 0.6913(+)   | 0.9227(=)   |
> | digen8_4426  | 0.5976 | 0.5384 | 0.6066(=)   | 0.5381(=)   |
> | digen9_7270  | 0.6306 | 0.6426 | 0.6292(=)   | 0.6461(=)   |
> | digen10_8322 | 0.6122 | 0.6732 | 0.6207(=)   | 0.6732(=)   |
> | +/=/-        | -      | -      | 2/8/0       | 1/9/0       |

---

> ### Author Response · Authors · 2022-11-15
> **Response to Reviewer bBog (Part II)**
>
> **Q2: In the real world, tabular data is usually very large-scale, and as the authors say in Section 5.1, they use part of the data due to limited computational resources. Therefore, an efficiency analysis (e.g., execution time, memory) of the proposed method is necessary, and a comparison with other baselines can also be added to comprehensively evaluate the applicability of each method.**
>
> A: Thanks. The execution time of different algorithms are presented in the supplementary material, and we also place them here. All the experiments are conducted on a single sever with identical CPU and RAM. Experimental results show that our method is faster than XGBoost, LightGBM and deep learning methods.
>
> | Algorithm     | Mean (Seconds) | Speedup |
> | :-: | :-: | :-: |
> | AutoFeat      | 16913.69       | 9.77x   |
> | FTTransformer | 13227.59       | 7.64x   |
> | ResNet        | 9973.16        | 5.76x   |
> | XGBoost       | 7073.36        | 4.09x   |
> | DCN           | 6984.05        | 4.03x   |
> | MLP           | 5821.68        | 3.36x   |
> | KNN           | 2223.24        | 1.28x   |
> | FEW           | 2087.66        | 1.21x   |
> | LightGBM      | 2023.95        | 1.17x   |
> | EvoFeat       | 1731.26        | 1.0x    |
> | RF            | 1348.87        | 0.78x   |
> | DT            | 1321.54        | 0.76x   |
> | LR            | 1010.16        | 0.58x   |
>
> Based on the comments from Reviewer LFDt, we add two additional baselines of increasing the upper limit of XGBoost and LightGBM to 5,000 estimators. The new experimental results are presented as follows. It is clear that our method is still competitive compared to these baselines.
>
> | Algorithm         |    Score | Improvement |
> | :-: | :-: | :-: |
> | EvoFeat           |            0.831899 | -                     |
> | LightGBM          |            0.823112 | 1.07%                |
> | LightGBM(5000)    |            0.821251 | 1.30%                |
> | XGBoost           |            0.819236 | 1.55%                |
> | XGBoost(5000)     |            0.813254 | 2.29%                |
> | RF                |            0.811038 | 2.57%                |
> | FEW(Default)      |            0.809664 | 2.75%                |
> | AutoFeat(Default) |            0.798939 | 4.13%                |
> | LightGBM(Default) |            0.796962 | 4.38%                |
> | XGBoost(Default)  |            0.796155 | 4.49%                |
> | FTTransformer     |            0.780356 | 6.60%                |
> | MLP               |            0.778617 | 6.84%                |
> | DT                |            0.775535 | 7.27%                |
> | DCN V2            |            0.772064 | 7.75%                |
> | ResNet            |            0.770875 | 7.92%                |
> | LR                |            0.760355 | 9.41%                |
> | KNN               |            0.735761 | 13.07%               |

---

> ### Author Response · Authors · 2022-11-15
> **Response to Reviewer bBog (Part III)**
>
> **Q3: The insights for the proposed method are not clear. It would be better to provide a more explicit motivation and vital support for the proposed approach than to beat some approaches only.**
>
> A: Thanks. The key motivation of this paper is to use a two-layer selection mechanism and the multitask optimization technique to evolve high-order features for a heterogeneous ensemble model. Based on an ensemble of decision trees and logistic regression models, our method is able to beat state-of-the-art machine learning algorithms. We add two experiments about the two-layer selection mechanism and the multitask optimization framework to demonstrate the validity of our contributions. Hoping these experiments can address your concerns.

---

### Official Review · Reviewer_8pAi · 2022-10-29

**Confidence:** 4
**Clarity, Quality, Novelty And Reproducibility:** The quality, clarity and originality …
**Correctness:** 3
**Technical Novelty And Significance:** 3
**Empirical Novelty And Significance:** 3
**Recommendation:** 6

**Strength And Weaknesses:**

Strengths:

1- This is a pioneering work that performs a search on features and learning models jointly (in the field of tabular data learning).

2- A solid number of experiments have been performed and the proposed algorithm outperforms various types of existing baseline models.

Weaknesses:

1- Some background information requires more detail (e.g. TGP, feature importance value). A more in-depth introduction to their design and their main characteristics will make the paper more throughout.

2- The influence of some design choices has not been analyzed empirically. For example, the design decision of using “self-competitive crossover” has not been analyzed.


**Summary Of The Paper:**

In this research, for the task of tabular data learning, an algorithm that searches for features used and the base learner used jointly has been proposed. Within the research, it has been found that on a large number of different tasks, the method outperforms the existing baselines.

**Summary Of The Review:**

Although there are some slight limitations in the paper, but the idea of “searching features and learning model jointly” is effective and the experiments are solid.

---

> ### Author Response · Authors · 2022-11-15
> **Response to Reviewer 8pAi (Part I)**
>
> **Q1: Some background information requires more detail (e.g. TGP, feature importance value). A more in-depth introduction to their design and their main characteristics will make the paper more throughout.**
>
> A: Thanks. Tree-based genetic programming (TGP) is a method focusing on evolving computer programs represented by symbolic trees. We use TGP because high-order features in machine learning can be naturally represented by symbolic trees. The details of the crossover and mutation operators in TGP are presented in the supplementary material.
>
> To calculate feature importance, for decision tree, EvoFeat uses the reduction of Gini impurity to calculate the importance value of each feature. For logistic regression, EvoFeat uses the normalized absolute value of the coefficient to determine the feature importance value of each feature. We compare these built-in feature importance calculation methods with SHAP values and the permutation importance method. Experimental results presented in the following table show that using built-in feature importance calculation method is significantly faster than other feature importance methods, while having similar effectiveness.
>
> |              | Built-in (Objective Values) | SHAP Values | Permutation Importance |
> | :-: | :-: | :-: | :-: |
> | digen1_6265  | 0.904      | 0.907 (=)     | 0.906 (=)                |
> | digen2_6949  | 0.51       | 0.5025 (=)    | 0.503 (=)                |
> | digen3_769   | 0.917      | 0.9195 (=)    | 0.921 (=)                |
> | digen4_860   | 0.557      | 0.553 (=)     | 0.5495 (=)               |
> | digen5_6949  | 0.6965     | 0.694 (=)     | 0.699 (=)                |
> | digen6_466   | 0.911      | 0.91 (=)      | 0.911 (=)                |
> | digen7_6949  | 0.545      | 0.5435 (=)    | 0.544 (=)                |
> | digen8_4426  | 0.5255     | 0.523 (=)     | 0.523 (=)                |
> | digen9_7270  | 0.6025     | 0.607 (=)     | 0.608 (=)                |
> | digen10_8322 | 0.6285     | 0.628 (=)     | 0.626 (=)                |
> | +/=/-        | -          | 0/10/0        | 0/10/0                   |
>
>
>
>
> |              |   Built-in (Time) | SHAP Values   | Permutation Importance   |
> |:-:|:-:|:-:|:-:|
> | digen1_6265  | 0.03       | 0.0365 (-)    | 0.1294 (-)               |
> | digen2_6949  | 0.0265     | 0.032 (-)     | 0.1166 (-)               |
> | digen3_769   | 0.029      | 0.036 (-)     | 0.1115 (-)               |
> | digen4_860   | 0.0248     | 0.03 (-)      | 0.1056 (-)               |
> | digen5_6949  | 0.025      | 0.0315 (-)    | 0.1066 (-)               |
> | digen6_466   | 0.027      | 0.033 (-)     | 0.1155 (-)               |
> | digen7_6949  | 0.028      | 0.032 (-)     | 0.1175 (-)               |
> | digen8_4426  | 0.027      | 0.036 (-)     | 0.1259 (-)               |
> | digen9_7270  | 0.026      | 0.0345 (-)    | 0.125 (-)                |
> | digen10_8322 | 0.0288     | 0.032 (-)     | 0.1138 (-)               |
> | +/=/-        | -          | 0/0/10        | 0/0/10                   |

---

> > ### Comment · Reviewer_8pAi · 2022-11-22
> > **Thank you for your answers**
> >
> > Thank you for your answers. By taking feedback from other reviewers into consideration, I will maintain my score unchanged.

---

> ### Author Response · Authors · 2022-11-15
> **Response to Reviewer 8pAi (Part II)**
>
> **Q2: The influence of some design choices has not been analyzed empirically. For example, the design decision of using “self-competitive crossover” has not been analyzed.**
>
> A: Thanks. We conduct ablation studies to show that combining self-competitive crossover (SC) with automatic lexicase selection (AL) can improve the performance of our task. Experimental results in the following table show that the AL+SC operator is the best choice.
>
> |Selection Operator| Mean of Accuracy | Improvement (EvoFeat) |
> | :-: | :-: | :-: |
> |AL+SC    | 0.860044|          - |
> |AL       | 0.857332|      0.32% |
> |SC       | 0.855892|      0.49% |
> |Random   | 0.841332|      2.22% |
>
> Moreover, using a two-layer selection mechanism based on feature importance values and cross-validation losses in a feature construction method is a key contribution of this paper. In order to illustrate why it is important to combine these two kinds of information, we add experiments in the supplementary material to test whether it is possible to use the random selection operator in each part.
>
> Experimental results are shown as below. It is clear that combing two kinds of selection operators (AL+SC) to leverage both feature importance values and cross-validation losses improves the performance of the automated feature construction method.
>
> |         | AL+SC   | AL+Random   | TN+Random   | TN+SC      | Random+Random   | Random+SC   |
> |:-:|:-:|:-:|:-:|:-:|:-:|:-:|
> | digen1  | 0.9065  | 0.879 (+)   | 0.881 (+)   | 0.8995 (+) | 0.836 (+)       | 0.86 (+)    |
> | digen2  | 0.5775  | 0.502 (+)   | 0.498 (+)   | 0.503 (+)  | 0.482 (+)       | 0.492 (+)   |
> | digen3  | 0.908   | 0.873 (+)   | 0.8685 (+)  | 0.8995 (+) | 0.823 (+)       | 0.856 (+)   |
> | digen4  | 0.791   | 0.523 (+)   | 0.517 (+)   | 0.524 (+)  | 0.501 (+)       | 0.505 (+)   |
> | digen5  | 0.6835  | 0.663 (+)   | 0.67 (+)    | 0.681 (=)  | 0.6485 (+)      | 0.659 (+)   |
> | digen6  | 0.539   | 0.529 (+)   | 0.532 (+)   | 0.539 (=)  | 0.52 (+)        | 0.521 (+)   |
> | digen7  | 0.54    | 0.5275 (+)  | 0.527 (+)   | 0.538 (=)  | 0.518 (+)       | 0.522 (+)   |
> | digen8  | 0.536   | 0.523 (+)   | 0.523 (+)   | 0.5335 (+) | 0.5115 (+)      | 0.515 (+)   |
> | digen9  | 0.608   | 0.6045 (+)  | 0.582 (+)   | 0.609 (=)  | 0.5415 (+)      | 0.544 (+)   |
> | digen10 | 0.6345  | 0.602 (+)   | 0.6025 (+)  | 0.636 (=)  | 0.561 (+)       | 0.584 (+)   |
> | +/=/-   | -       | 10/0/0      | 10/0/0      | 5/5/0      | 10/0/0          | 10/0/0      |

---

### Author Response · Authors · 2022-11-15
**General Response**

First of all, we would like to express our gratitude to reviewers for their thorough reading of the manuscript and insightful comments. We have revised our paper and our major changes are as follows:

**Expansion of Search Space for XGBoost:** Reviewers bBog and LFDt raised concerns about whether our comparisons are fair. We have tried our best to make fair comparisons. For example, we use the state-of-the-art Bayesian optimization framework (HEBO) to tune hyperparameters of XGBoost and LightGBM to increase the chance of finding the optimal hyperparameters in limited time. Also, we add more experiments to expand the hyperparameter search space of XGBoost and LightGBM by allowing them to have up to 5000 trees according to the suggestion from Reviewer LFDt.

**Analysis of Key Component:** Reviewers 8pAi and bBog raised concerns about the contributions of the key components. We add two experiments in the supplementary material to highlight our contributions and show the validity of our contributions about the two-layer selection mechanism and the multitask optimization framework. Also, we add an experiment to prove the effectiveness of using the built-in feature importance calculation method for calculating feature importance values.

---

### Decision · Program_Chairs · 2023-01-20

**Decision:**

Reject

**Justification For Why Not Higher Score:**

The authors may have overclaimed the significance of the proposed study here. The authors tried to explore the case with heterogeneous features and a limited amount of data, but that is not the scenario for ordinary deep learning. The authors proposed an ad-hoc approach to select or design features for the classifiers, but they did not take the literature on feature selection, metric learning, or dimension reduction into account, which makes the current experimental design/comparison incomplete or unfair.

**Justification For Why Not Lower Score:**

N/A

**Metareview: Summary, Strengths And Weaknesses:**

This paper introduced an evolutionary algorithm to do feature engineering, while the neural network based features can be easy to overfitting. The proposed algorithm has been clearly introduced and the authors have compared the proposed algorithm with various baselines (e.g., ML-based, DL-based, etc). But its limited technical novelty could be a major weakness.